# Impact of work competencies on job performance among university counsellors

Jie Cao[1], Nur Naha Abu Mansor[2], Jinhua Li[3]*

1 Faculty of Education, Xiangnan University, Chenzhou, Hunan Province, China, 2 Faculty of Business, Sohar University, Sohar, Oman, 3 Faculty of Arts and Design, Xiangnan University, Chenzhou, Hunan Province, China

* lijinghua@xnu.edu.cn

**Data Availability Statement:** All relevant data are within its Supporting Information files.

**Funding:** The author(s) received no specific funding for this work.

## Abstract

This study aims to examine the relationship between work competency factors—namely knowledge, skills, traits, motives, and self-concepts—and job performance among university counsellors in China. Data were collected from 310 university counsellors using multiple sampling techniques. Partial Least Squares Structural Equation Modeling (PLS-SEM), via Smart-PLS version 3.3.2, was employed for data analysis. The findings revealed that four competency factors—knowledge, skills, traits, and motives—were significantly related to job performance among university counsellors in Hunan Province. These results contribute to a deeper understanding of the impact of work competencies on job performance. It can provide valuable insights for administrators and policymakers aiming to improve the competencies and performance of university counsellors, as well as to enhance the overall development of the counselling team.

## Introduction

The university counsellor is a unique role within Chinese universities, responsible for managing student affairs. In 2006, the Ministry of Education of the People's Republic of China issued the "Provisions on the Construction of Counsellors in Ordinary Universities," which formally defined the role of university counsellors at a national level. This regulation was updated in 2017, positioning university counsellors as integral members of both the teaching and student management teams. It recognized counsellors as the backbone of ideological and political education for university students, as well as the organizers, implementers, and conductors of students' daily ideological and political education and management [1]. The regulation also outlined specific work requirements and responsibilities, providing a foundation for the selection, training, and employment of university counsellors.

To some extent, university counsellors in China resemble student affairs administrators in other countries, but there are notable differences in the roles, professional development and management mechanisms [2]. In China, university counsellors hold a management position and are considered a type of administrator within the university system. Counsellors are selected through a combination of organizational recommendations and open recruitment, which involves written examinations, interviews, and other procedures [1]. Unlike student

**Competing interests:** The authors have declared that no competing interests exist.

affairs administrators in Western countries, who are often required to have relevant professional qualifications and experience, university counsellors in China need only a bachelor's degree or higher, without specific professional qualification requirements [3, 4]. For example, in the United States, student affairs administrators must possess specialized skills and qualifications, often holding degrees in related fields and gaining experience in student affairs management [2].

There is no denying that university counsellors play a critical role in educating, guiding, nurturing, and shaping students. Their abilities and qualities directly impact their competency in performing their duties, determining whether they can meet the standards of being politically strong, professionally competent, and of a correct style. This, in turn, has a significant influence on the healthy growth and holistic development of university students [5]. As noted by Ding, the competencies of counsellors are crucial for the effective execution of university students' ideological and political education, employment services, public opinion supervision, and daily management [6].

However, with social and economic development, China's higher education system has entered an era of mass education, leading to a significant increase in the number of university students. The values of contemporary students have also diversified: students born in the 1990s and 2000s have different personality traits and educational needs. Additionally, the widespread use of the internet has introduced complex changes to the cultural environment in which students live [7]. These changes affect students' values, moral sentiments, political beliefs, and ideals, making ideological education increasingly challenging [8]. Therefore, the competency level of university counsellors must be further improved in order to face the changes.

Previous studies have shown that individuals' work competencies significantly influence their job performance [9–12]. However, research on university counsellors remains insufficient. One notable gap is the evolving nature of the profession. As society advances and technology progresses, the form and requirements of education inevitably change, creating new demands on university counsellors. Counsellors today face numerous challenges, such as managing complex student issues, handling diverse responsibilities, and continuously improving their own competencies [13, 14].

Currently, the work competencies of university counsellors in China are not particularly optimistic due to uneven staffing, unreasonable knowledge structures, and other factors [15–18]. Liang pointed out that improving the overall competencies of university counsellors is a core issue in strengthening the management and education of university students [13]. Similarly, Song identified several notable problems among counsellors, such as poor stress management, inadequate moral and ideological education, and low job loyalty, leading to generally low competency levels [19]. Without adequate competencies, counsellors struggle to manage student affairs effectively, which further highlights the need to examine counsellors' competencies. Additionally, there is no standardized system for evaluating their job performance [20, 21]. The complexity of their work also complicates the development of assessment indicators, making their performance difficult to measure [22–24].

Another gap in the research is the limited focus on work competencies and job performance in the education sector. While much attention has been given to these topics in industries like business, government, and healthcare [23, 25–27], few studies have explored the competencies and job performance of academic administrators, particularly university counsellors in China [28–30].

Lastly, previous research on the relationship between competencies and job performance has yielded inconsistent findings. Some studies suggest that employees with high competencies can adapt quickly to workplace changes, which positively affects their performance [10, 11,

31]. However, other studies report a negative relationship between competencies and job performance [32, 33]. Furthermore, little research has examined the influence of specific competency facets—such as knowledge, skills, traits, motives, and self-concepts—on job performance [34, 35]. Therefore, this study seeks to investigate whether these facets of work competency affect job performance in the workplace.

### Literature review and hypothese development

**Work competencies.** Work competencies refer to the behavioral aspects of an individual's ability to perform a job competently, encompassing virtually anything that can directly or indirectly impact job performance [36]. They represent the fundamental knowledge and performance criteria necessary to successfully perform in a role or qualify for a position [37]. Many modern organizations develop competency frameworks and work standards tailored to the specific requirements of each role, serving as a basis for employee selection, training, and evaluation. This approach is equally relevant in the education sector. Introducing competency research to university counsellors is of significant importance, as it helps clarify their role, deepens their professional identity, and enables counsellors to enhance their job performance by improving their skills and work methods [38–40].

Spencer and Spencer define competencies as an "underlying characteristic of an individual that is causally related to criterion referenced effective and/or superior performance in a job or situation" [41]. These underlying characteristics encompass motives, traits, self-image, attitudes, values, knowledge in a particular field, and skills, all of which can be reliably measured and distinguished between high and average performance [41–43]. Hoffmann further clarified that "competency" refers to behavior, while "competence" refers to standards [44]. At the same time, Shi considered that competencies referred to behavior types and psychological nature which were causally related to excellent performance, while competence referred to what must be done and its standards [45]. Despite these distinctions, many scholars believe that the terms "competency" and "competence" are often used interchangeably in academic discourse today [46, 47].

Various scholars have offered similar explanations of university counsellors' work competencies, drawing upon Spencer's conceptual framework. Zhao, for example, suggested that the work competency of university counsellors encompasses the qualities, traits, knowledge, skills, and other personality characteristics that directly affect job performance and promote the holistic development and success of students [48]. Yang, on the other hand, noted that counsellors' competencies are not only a collection of the characteristics required for the job but also reflect students' expectations of their role [40]. Based on the universities' orientations, training objectives, and counsellors' duties, these competencies represent a combination of knowledge, skills, abilities, and other personality traits that are integral to the effective management of student affairs. Ultimately, these competencies contribute to the achievement of universities' educational goals and foster the healthy development of students [40].

In summary, the work competencies of a university counsellor include the knowledge they must acquire, the skills they need to develop, the traits suited for the position, the motives for choosing the profession, and the self-concept related to the job. Specifically, knowledge is defined as professional information needed to complete the tasks of counsellors and is necessary to facilitate university counsellors in performing their job roles in this research. The professional knowledge includes psychological knowledge, pedagogical knowledge, management knowledge, career planning knowledge, ideological and political knowledge, professional knowledge related to student, party and league knowledge. Skills represent all the abilities that counsellors need to master. These specialised skills can be applied to efficiently complete their

daily work tasks and perform well, including learning skills, innovation skills, communication skills, crisis management skills, organisation skills, psychological counselling skills, analytical judgment, interpersonal skills, information collection, transposition thinking, identification, insight and psychological adjustment ability. traits refer to university counsellors' own qualities or characteristics, which will influence their working patterns. Self-concept is the social image that counsellors want to build in their work, such as being a psychologist, an educator, a life assistant, and so on. It is also the understanding of their own ability and self-value. Motives are the incentives of counsellors to choose this job, which mainly include needs, belongings, importance, interests, ideals and beliefs, respect, and job identity. All these competencies can effectively predict the performance of university counsellors.

**Job performance.** Job performance is widely recognized as a critical component of organizational management, with its measurement and standards playing a key role in the effectiveness of an organization [49]. Typically, organizations evaluate individual employees' contributions through job performance metrics and establish fair reward and penalty systems based on these evaluations [50]. As a result, job performance has garnered significant attention from scholars and professionals both domestically and internationally.

Campbell defined job performance as the behavior exhibited by employees or organizational members in fulfilling the tasks assigned by the enterprise or organization [51]. According to Campbell and colleagues, performance is characterized by specific behaviors that are goal-oriented, and they assert that "performance is not the consequence or result of action, it is the action itself" [52]. This view aligns with Motowidlo et al., who also posited that performance is a behavioral phenomenon that is directly related to an organization's objectives and can be measured by the extent to which employees contribute to those objectives [53]. Job performance emphasizes the comparison between an employee's actual performance and the organization's expected contribution from its employees, offering measurable and quantifiable standards to assess the value of individual performance.

**Hypotheses development.** From the definition of competencies, most scholars agree that competencies are individual characteristics that influence performance and can differentiate efficient individuals from inefficient ones [40, 41, 45, 47, 54–58].

For example, Nasir et al. conducted quantitative research on lecturers' performance in Indonesia and demonstrated that lecturers' competency had a significant impact on their job performance, with a path coefficient of 0.307 [10]. This finding aligns with Sukrapi et al., who found a significant positive correlation between teachers' work competencies and job performance, suggesting that teachers with higher competencies achieve better performance [59].

However, Sumantri and Whardani presented a different view, arguing that some lecturers with lower competencies can still perform well, while others with higher competencies may struggle to achieve high performance [9]. This was in line with the findings of Chen [60]. Chen demonstrated that the sale competency of car salespeople had no significant correlation on the job performance [60]. Conversely, Liang's study on counsellors found a significant positive correlation between the sub-dimensions of counsellors' competencies and job performance [13]. The outcome was also compatible with the finding of Y. Yang [38] and Yang et al. [61].

Similarly, Peng stated that strengthening counsellors' competencies could improve their job performance [62]. Xu and Ye considered that competency can be one of the performance determinants [63]. Other studies have consistently shown that employees who possess the necessary competencies for their roles can improve their job performance [13, 18, 41, 55, 61, 64]. However, Song indicated that cognitive competency had no significant correlation with job performance, while other competencies such as knowledge, skills, virtues, and interpersonal interactions were statistically significant predictors [65].

Kiatsuranon and Suwunnamek held the opinion that employee competency was directly related to organization performance in the ICT industry [11]. This was consistent with the point of Arifin, who proved that teachers' competency can positively affect the job performance, which meant that teachers with high competency can achieve high performance [66]. Nevertheless, Winarno and Perdana also stated that high competency will not deliver elevated performance without the assistance of high motivation [32]. Overall, competencies are critical for enhancing job performance across various industries.

Numerous studies have confirmed the strong relationship between job competencies and performance [12, 32, 67–69]. Thus, the five dimensions of work competencies may also be correlated to job performance.

Knowledge, in particular, is seen as a crucial factor in achieving competitive advantage and high performance in organizations [70]. Liu et al. argued that returnee entrepreneurs' business knowledge from abroad gives their companies competitive advantages, thus improving organizational performance [71]. Rangchian et al.'s study in the Iranian pharmacy industry demonstrated that enhanced knowledge, skills, and behavior positively impacted pharmacy performance [72]. Similarly, Groza and Groza found that salespeople's regulatory knowledge was significantly related to sales performance [73]. Sujatha and Krishnaveni emphasized that knowledge improves performance through product and process development [74]. Liu and Guo noted that professional knowledge significantly affects university counsellors' performance, making it an important criterion for recruitment, promotion, and training [75].

Previous study displayed that skills were closely related to job performance [76–78]. Sudi et al. found that the communication skills can affect the administrative performance in higher education institutions [77]. This is in line with the research of Moghimi et al., who stated that employees with stronger communication skills can affect their performance as it can support them achieve personal and organization goals [79]. Meanwhile, Cayır and Ulupınar highlighted that improved educational skills lead to better performance among nurse instructors [78]. For university counsellors, Chen noted that professional skills can enhance work outcomes [80]. This is consistent with the research of Lei, who stressed that improving counsellors' professional skills leads to better performance [81].

The relationship between personality traits and job performance has been a longstanding area of research [82]. Prior to the 1990s, the link between traits and performance was unclear [83–85]. It was in 1991 when the relationship between personality and job performance was clear and more convinced [86]. Tett and Jackson confirmed that traits, particularly those in the five-factor model, are strongly related to job performance [87]. Subsequent studies by Salgado [88] and Hurtz and Donovan [89] supported these findings.

Meanwhile, other researchers also held their own opinion about the two variables: traits and job performance. Motowidlo et al. further argued that traits like conscientiousness affect task performance by influencing task habits [53], while Behling emphasized that traits are crucial for job performance [90]. Oppong et al. thought that because of neglecting the importance of traits, many institutions were suffering the situation of dissatisfaction with job and low employee performance [91]. Ghani et al. demonstrated that leaders' personality traits significantly impact their job performance [92].

Research on motives has also explored their relationship with work behavior and performance. Sharma and Sharma found that personal incentives motivate employees to work harder, thus improving performance [93]. Moreover, Afful-broni also concluded that low salaries and lack of motives will reduce performance in universities after doing research in a selected university in Tarkwa [94]. Zlate and Cucui noted that motivation in higher education directly impacts staff performance, highlighting the need for institutions to foster motivational

mechanisms [95]. Ren's research indicated that achievement motives positively predict job performance [96].

The link between self-concept and job performance remains debated. Khalaila's research showed a strong correlation between students' self-concept and academic performance [97]. Sikhwari also held the same opinion, there was a significant correlation between self-concept and academic performance, and these two variables mutually impacted and determined each other [98]. Christoph et al. demonstrated that computer self-concept influences performance [99], while Dockx et al. offered a different view, suggesting that students in high-performing classes may develop lower academic self-concept [100]. Otherwise, some researchers held the opposite opinion from Dockx et al. [100]. Albert and Dahling considered that self-concept can be a critical predictor of academic job performance among students [101]. Liao emphasized the importance of self-esteem in driving high performance [102]. This is in line with Zanden et al., who emphasized that self-concept and performance were complementary and interrelated, so the measures aiming at improving performance should simultaneously reinforce self-concept [103].

In summary, although research on work competencies has yielded fruitful results, there remain differing views on the relationship between competency and job performance. Some scholars argue that competencies significantly influence performance across various fields, while others present contrary perspectives. These inconsistencies highlight the need for further investigation, particularly into the influence of competency dimensions on job performance. Based on the literature reviewed, the following hypotheses are proposed. Fig 1 displays the conceptual framework.

H1: Knowledge will be positively related to job performance among university counsellors in Hunan province.

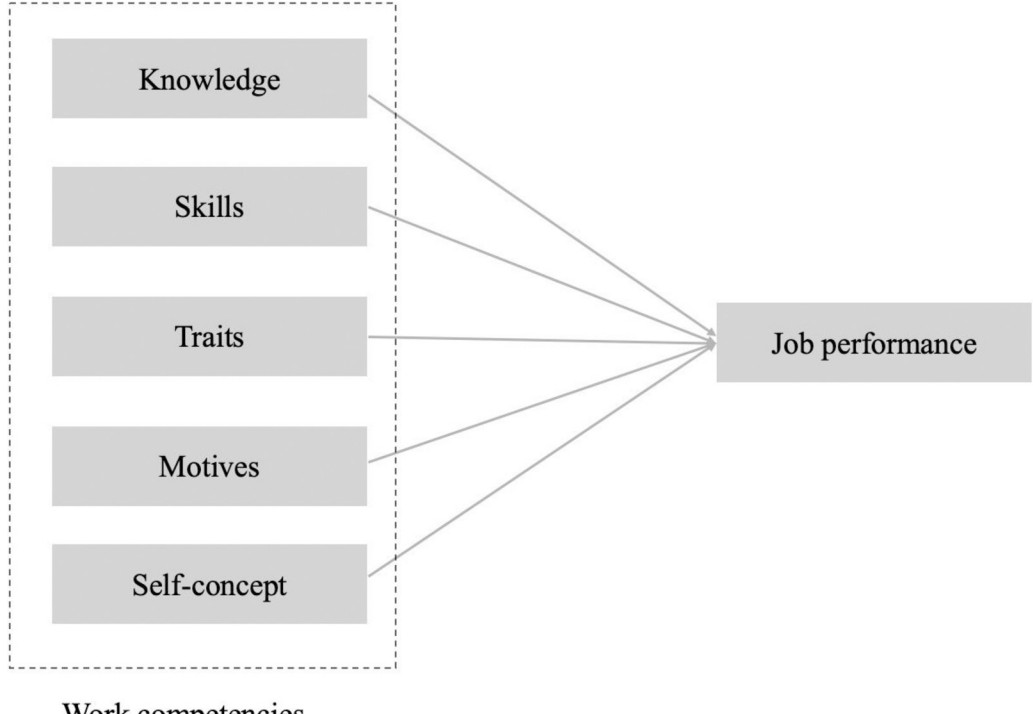

**Fig 1. Conceptual framework.**

H2: Skills will be positively related to job performance among university counsellors in Hunan province.

H3: Traits will be positively related to job performance among university counsellors in Hunan province.

H4: Motives will be positively related to job performance among university counsellors in Hunan province.

H5: Self-concept will be positively related to job performance among university counsellors in Hunan province.

## Theoretical foundation

This research investigates the impact of work competencies on job performance among university counsellors. It examines six constructs: knowledge, skills, traits, motives, self-concepts, and job performance. The relationships between these variables are explained through competency theory, which serves as the theoretical foundation of the study. Competency theory posits that counsellors' job performance can be enhanced by improving their work competencies. Developed by Spencer and Spencer, competency theory focuses on the underlying characteristics of employees that influence performance, distinguishing exceptional individuals from average performers [41]. Spencer and Spencer highlighted that these underlying characteristics, referred to as competencies, are crucial for achieving superior performance in the workplace [41]. Competencies are deeply embedded in an individual's personality and can predict a wide range of behaviors and performance outcomes. These characteristics typically include knowledge, skills, traits, self-concepts, and motives [34, 41, 104].

Knowledge refers to the specialised information required within a certain professional field [41, 56, 104]. Skills involve the ability to master and apply specialized techniques to complete tasks [41, 104]. Traits refer to an individual's patterns of reaction, tendencies, and characteristics when interacting with external environments and processing various kinds of information [41, 104]. Motives refer to the internal driving force that promote and maintain individuals to take actions to achieve certain goals [41, 104, 105]. Self-concept refers to a person's attitudes, values, or self-image [41, 104].

Competencies such as knowledge and skills are considered threshold competencies—those basic qualities that are visible and essential for competent performance. They represent the minimum requirements for effectively completing work tasks but do not distinguish exceptional performers from average ones [41]. These are surface-level competencies, often compared to the part of an iceberg visible above water. Conversely, self-concept, traits, and motives represent differentiating competencies. These are deeper, often hidden characteristics that are not immediately perceivable but are key factors determining behavior and performance [41]. These deeper competencies are more influential in distinguishing superior performers from the rest. Thus, the five dimensions of work competencies—knowledge, skills, traits, motives, and self-concept—are essential in shaping job performance. The competency model serves as the framework of this study, illustrating the relationship between these competencies and counsellors' job performance.

## Research method

This study distributed questionnaires alongside the written informed consent to the respondents from Novermber 15 until December 30 in 2023. The informed consent is mentioned in the introduction of the questionnaire, including the purpose and significance of the study, the

research object, the data use, etc. If the respondents choose agree in the informed consent option, they are deemed to have agreed to participate in this survey and begin to fill in the subsequent questionnaire. This research was approved by the Ethics Committee of Xiangnan University with ethics approval reference 2023YXLL045.

## Instruments

**Work competencies.** Work competency questionnaire was adapted from the research of J. Jin et al. [106], Chen [47] and Lv [16]. It consists of 42 questions across five scales: knowledge (7 items), skills (13 items), self-concept (6 items), traits (10 items), and motives (6 items). In order to keep the consistency of the questionnaire and increase the understanding of respondents, the subject "I" was added before the questions. Sample items include "I have the basic psychological knowledge", "I have the ability of communicate, which can communicate with others effectively", "I have a close friendship with students", "I have full enthusiasm for work", "I can get a sense of belongings from this job". The Cronbach's alpha value of the scale was 0.965.

**Jop performance.** Job performance instrument was adapted from Ma [50] to measure university counsellors' job performance. The scale includes 27 items, with "you" in the original questions replaced by "I" for consistency. A sample item is: "I have certain work plans about this work". The Cronbach's alpha value of the scale was 0.951.

## Sample

This research focused on public universities in the second batch category in Hunan province. These universities are typically organized by provincial, regional, or city-level governments and receive local financial support, prioritizing the training of local talent and serving regional industries. However, these local universities face numerous challenges in developing their counsellors, such as unbalanced structures, multi-role expectations, unclear job responsibilities, staff shortages, and inadequate evaluation and incentive mechanisms [107]. Public universities were selected for this study because private universities are relatively few in Hunan province. Although private universities in China are funded by non-state financial sources and managed by social organizations or individuals, they face additional risks, including financial, quality, management, policy, and market risks [108]. Due to operational constraints, student demographics, and social standing, private universities are often seen as more vulnerable in China's higher education system [109]. The mainstream of higher education is still public universities in China. Thus, this study only chose public university to research.

The total number of counsellors in public universities among second batch was 667. Thus, the target population of this research was 667. All the data came from the Ideological and Political Department of Hunan Education Department, which is the department in charge of university counsellors throughout the whole province. Thus, it can ensure the authenticity of the data.

According to sample size calculation method of Chua [110], the final sample size of this study was 310 university counsellors. A proportionate stratified sampling method was employed, with 75 respondents from the capital city and 235 respondents from other cities within Hunan province. To select the participants, simple random sampling was applied [111]. Data collection took place from November 15 to December 30, 2023, with questionnaires distributed alongside informed consent. After data cleaning, 13 respondents were excluded, leaving a final sample of 297 counsellors for analysis. Table 1 displays the demographic information of the respondents.

**Table 1. Demographic information.**

| Variable | Category | Frequency | Percent |
|---|---|---|---|
| Gender | Male | 125 | 42.1 |
| | Female | 172 | 57.9 |
| Age | 25 years and less | 50 | 16.8 |
| | 26–35 years old | 183 | 61.6 |
| | 36–45 years old | 53 | 17.8 |
| | 46–55 years old | 7 | 2.4 |
| | 56 years and above | 4 | 1.3 |
| Marital status | Single | 122 | 41.1 |
| | Married | 169 | 56.9 |
| | Divorce | 6 | 2 |
| Education | College degree | 2 | 0.7 |
| | Bachelor's degree | 70 | 23.6 |
| | Master's degree | 218 | 73.4 |
| | Doctor degree | 7 | 2.4 |
| Working experience | 1–5 years | 185 | 62.3 |
| | 6–10 years | 58 | 19.5 |
| | 11–15 years | 20 | 6.7 |
| | 16–20 years | 20 | 6.7 |
| | 21–25 years | 5 | 1.7 |
| | 26 years and above | 9 | 3.0 |
| Title | Primary title | 192 | 64.6 |
| | Middle title | 91 | 30.6 |
| | Vice-senior title | 11 | 3.7 |
| | Senior title | 3 | 1 |
| Income | Less than RMB 2000 | 7 | 2.4 |
| | RMB2001-3500 | 26 | 8.8 |
| | RMB3501-4000 | 75 | 25.3 |
| | RMB4001-4500 | 53 | 17.8 |
| | RMB4501-5000 | 57 | 19.2 |
| | More than RMB 5000 | 79 | 26.6 |

## Data analysis and results

In this study, the researcher employed the PLS-SEM technique for analysis. One of the primary reasons for selecting this method was the nature of the research model, which aimed to examine the relationship between work competencies and job performance among counsellors. While the model was supported by existing theoretical frameworks, there was a lack of empirical studies to substantiate it. As noted by Hair et al. [112, 113], PLS-SEM is well-suited for theory extension and exploratory research, making it an appropriate choice for this study.

Furthermore, Hair et al. [114] highlighted that PLS-SEM is capable of testing predictive relationships among constructs. In this study, the method was used to predict the influence of each variable on the others. Moreover, PLS-SEM is effective in handling complex models, accommodating wider sample sizes, and dealing with fewer data constraints, thus enabling researchers to address a broader range of research problems [115].

The statistical tool used for data analysis in this study was SmartPLS version 3.3.2 [116]. There are two stages of model evaluation in PLS, that is measurement model assessment and structural model assessment [113, 117, 118]. The measurement model assessment includes the

internal consistency reliability, convergent validity, and discriminant validity [113, 117]. The structural model assessment involves six key steps: evaluating the structural model for collinearity issues, assessing the significance and relevance of structural model relationships, examining the $R^2$ value, determining the effect size ($f^2$), assessing predictive relevance ($Q^2$), and optionally, assessing the $q^2$ effect size [117, 119].

Self-reported questionnaires were employed to collect data for this research. Since both the independent and dependent variables were obtained from the same respondents, it was crucial to assess for common method bias [120]. Common method variance (CMV) efers to the systematic error variance among variables, caused by the use of similar measurement tools rather than reflecting the true relationships between the constructs [121]. he more homogeneous the data source and measurement methods, the greater the risk of CMV affecting the validity of the research findings and introducing bias [122]. This is particularly relevant when a single source and the same respondents are used for all the measurements [122].

Researchers can mitigate the potential impact of CMV through procedural and statistical remedies [120]. This study used Harman's single-factor test to check the bias of common methods. In this study, Harman's single-factor test was conducted to assess common method bias. All variables were loaded into an exploratory factor analysis, and the unrotated factor solutions were examined to determine the number of factors required to explain the variance. If only one factor emerged, or if a single factor accounted for the majority of the covariance among the measures, it would indicate the presence of common method variance [120]. The result showed that the total variance explained by the first factor was 37.176%, and more than one factor was present, indicating that common method variance was not a significant issue in this study.

## Measurement model

In this study, composite reliability (CR) was used to measure internal consistency reliability, rather than the traditional Cronbach's alpha (α), as Cronbach's alpha tends to underestimate true reliability [117, 123, 124]. The recommended threshold for composite reliability is a value greater than 0.7 [119]. As shown in Table 1, all constructs demonstrated composite reliability values exceeding 0.7. Job performance had the highest CR value (0.962), followed by traits (0.939), skills (0.937), motives (0.914), and self-concept (0.896). These results indicate that all CR values met the 0.7 threshold, demonstrating acceptable internal consistency reliability for the measurement model.

Convergent validity refers to the extent to which different measurement methods yield consistent results when assessing the same construct. In other words, various measurement approaches should converge when measuring the same construct [117, 119]. Convergent validity is typically evaluated using outer loadings and the average variance extracted (AVE) [117, 119]. Outer loadings assess the degree to which an indicator aligns with the construct it is intended to measure [125], with a recommended threshold of 0.5 or higher [126]. Indicators with outer loadings below 0.5 are suggested for removal. In this study, after running the PLS algorithm, PER 14 and PER 27 were deleted as their outer loadings did not meet the 0.5 threshold. Another assessment for convergent validity was AVE, representing the average of the outer loadings across all indicators related to a construct [117]. The cut-off value for AVE is 0.5, meaning that each construct should explain at least 50% of the variance of its indicators to demonstrate convergent validity [119]. As seen in Table 2, all AVE values exceeded the 0.5 threshold, indicating that the measurements were both reliable and valid.

Discriminant validity assesses the extent to which a construct is empirically distinct from other constructs [119]. In this study, discriminant validity was evaluated using the heterotrait-

**Table 2. Measurement model.**

| Constructs | Items | Loadings | CR | AVE |
|---|---|---|---|---|
| Con | CON1 | 0.760 | 0.896 | 0.59 |
| | CON2 | 0.744 | | |
| | CON3 | 0.706 | | |
| | CON4 | 0.733 | | |
| | CON5 | 0.808 | | |
| | CON6 | 0.848 | | |
| KNO | KNO1 | 0.678 | 0.899 | 0.561 |
| | KNO2 | 0.704 | | |
| | KNO3 | 0.771 | | |
| | KNO4 | 0.792 | | |
| | KNO5 | 0.780 | | |
| | KNO6 | 0.767 | | |
| | KNO7 | 0.742 | | |
| MOT | MOT1 | 0.848 | 0.914 | 0.641 |
| | MOT2 | 0.847 | | |
| | MOT3 | 0.841 | | |
| | MOT4 | 0.662 | | |
| | MOT5 | 0.818 | | |
| | MOT6 | 0.772 | | |
| PER | PER1 | 0.704 | 0.962 | 0.507 |
| | PER10 | 0.796 | | |
| | PER11 | 0.672 | | |
| | PER12 | 0.736 | | |
| | PER13 | 0.701 | | |
| | PER15 | 0.618 | | |
| | PER16 | 0.649 | | |
| | PER17 | 0.587 | | |
| | PER18 | 0.652 | | |
| | PER19 | 0.630 | | |
| | PER2 | 0.703 | | |
| | PER20 | 0.701 | | |
| | PER21 | 0.767 | | |
| | PER22 | 0.781 | | |
| | PER23 | 0.791 | | |
| | PER24 | 0.688 | | |
| | PER25 | 0.702 | | |
| | PER26 | 0.724 | | |
| | PER3 | 0.696 | | |
| | PER4 | 0.757 | | |
| | PER5 | 0.752 | | |
| | PER6 | 0.763 | | |
| | PER7 | 0.726 | | |
| | PER8 | 0.727 | | |
| | PER9 | 0.730 | | |
| SKI | SKI1 | 0.711 | 0.937 | 0.534 |
| | SKI10 | 0.749 | | |
| | SKI11 | 0.709 | | |

(*Continued*)

**Table 2.** (Continued)

| Constructs | Items | Loadings | CR | AVE |
|---|---|---|---|---|
| | SKI12 | 0.763 | | |
| | SKI13 | 0.695 | | |
| | SKI2 | 0.780 | | |
| | SKI3 | 0.647 | | |
| | SKI4 | 0.787 | | |
| | SKI5 | 0.722 | | |
| | SKI6 | 0.743 | | |
| | SKI7 | 0.772 | | |
| | SKI8 | 0.716 | | |
| | SKI9 | 0.690 | | |
| TRA | TRA1 | 0.620 | 0.939 | 0.608 |
| | TRA10 | 0.772 | | |
| | TRA2 | 0.761 | | |
| | TRA3 | 0.847 | | |
| | TRA4 | 0.798 | | |
| | TRA5 | 0.797 | | |
| | TRA6 | 0.753 | | |
| | TRA7 | 0.811 | | |
| | TRA8 | 0.801 | | |
| | TRA9 | 0.814 | | |

Notes: KNO: knowledge; SKI: skills; TRA: traits; MOT: motives; CON: self-concept; PER: job performance. PER 14 and PER 27 were deleted because of low loadings

monotrait ratio (HTMT) [127]. According to Hair et al. [119], HTMT was the ratio of between-trait and within-trait. It was the ratio of the mean value of indicators correlation between different constructs to the mean value of indicators correlation within the constructs [117, 119]. HTMT can be used as a criterion and a statistical test when assessing discriminate validity. HTMT value lower than 0.90 indicated that there was no problem of discriminant validity when using it as a criterion [128]. When regarding it as a statistical test, HTMT inference was assessed. The HTMT confidence interval did not include the value of 1 demonstrated that the discriminant validity was achieved. Table 3 presents the results of discriminant

**Table 3. Discriminant validity.**

| | CON | KNO | MOT | PER | SKI | TRA |
|---|---|---|---|---|---|---|
| **CON** | | | | | | |
| **KNO** | 0.658 CI.90(0.551,0.738) | | | | | |
| **MOT** | 0.699 CI.90(0.618,0.767) | 0.513 CI.90(0.411,0.602) | | | | |
| **PER** | 0.815 CI.90(0.747,0.867) | 0.777 CI.90(0.715,0.829) | 0.746 CI.90(0.672,0.810) | | | |
| **SKI** | 0.899 CI.90(0.861,0.933) | 0.792 CI.90(0.723,0.842) | 0.672 CI.90(0.586,0.753) | 0.921 CI.90(0.889,0.944) | | |
| **TRA** | 0.904 CI.90(0.857,0.947) | 0.543 CI.90(0.444,0.638) | 0.743 CI.90(0.672,0.806) | 0.799 CI.90(0.740,0.851) | 0.756 CI.90(0.693,0.813) | |

Notes: KNO: knowledge; SKI: skills; TRA: traits; MOT: motives; CON: self-concept; PER: job performance

**Table 4. The result of inner VIF values.**

|  | PER | CON | KNO | MOT | SKI | TRA |
|---|---|---|---|---|---|---|
| PER |  |  |  |  |  |  |
| CON | 4.328 |  |  |  |  |  |
| KON | 2.046 |  |  |  |  |  |
| MOT | 1.986 |  |  |  |  |  |
| SKI | 4.258 |  |  |  |  |  |
| TRA | 3.418 |  |  |  |  |  |

Notes: KNO: knowledge; SKI: skills; TRA: traits; MOT: motives; CON: self-concept; PER: job performance

validity. Although two HTMT values exceeded the 0.90 threshold, a bootstrapping procedure was conducted to examine the HTMT confidence intervals. The results showed that the confidence intervals did not include the value of 1, confirming discriminant validity. Based on the guidelines of Henseler et al. [127], Franke and Sarstedt [129] and Ramayah et al. [117], the measurement model exhibited sufficient discriminant validity.

## Structural model

After assessing the measurement model, the structural model was examined to evaluate its predictive power and the relationships between the constructs [119], which directly addressed the research objectives of this study.

The value of path coefficient, standard error, p-value, t-value, confidence intervals and effect sizes were all reported in this structural model through conducting bootstrapping procedure with 5000 resamples and one-tail set. The threshold of t-values for one-tails test were 2.33 for significance level of 1 percent, 1.645 for significance level of 5 percent, and 1.28 for significance level of 10 percent.

While vertical collinearity may be resolved, lateral collinearity could still pose an issue [117]. Therefore, it was important to evaluate collinearity for each subset of the structural model by assessing the predictor constructs individually [130]. To detect potential collinearity issues, the variance inflation factor (VIF) was used in the analysis. According to Hair et al. [115], a VIF value exceeding 5 indicates a potential collinearity problem. As shown in Table 4, the inner VIF values for all independent variables were below 5, confirming that collinearity was not a concern in this study.

After running the PLS algorithm in SmartPLS, the path coefficients representing the relationships between constructs were obtained. The values of path coefficients range from -1 to +1, with a coefficient close to +1 indicating a strong positive relationship, a value near -1 signifying a strong negative relationship, and a value close to 0 suggesting a weak relationship [117, 119]. However, the significance of a coefficient depends on its standard error, which is

**Table 5. Hypothesis testing the direct relationship.**

| Hypothesis | Relationship | Std Beta | Std Error | T-value | P-value | BCI LL | BCI UL | $f^2$ |
|---|---|---|---|---|---|---|---|---|
| H1 | KNO -> PER | 0.188 | 0.043 | 4.398 | P < .01 | 0.114 | 0.273 | 0.105 |
| H2 | SKI-> PER | 0.526 | 0.060 | 8.806 | P < .01 | 0.403 | 0.645 | 0.394 |
| H3 | TRA -> PER | 0.260 | 0.055 | 4.738 | P < .01 | 0.159 | 0.372 | 0.120 |
| H4 | MOT -> PER | 0.173 | 0.042 | 4.078 | P < .01 | 0.088 | 0.253 | 0.092 |
| H5 | CON-> PER | -0.100 | 0.059 | 1.703 | 0.089 | -0.215 | 0.018 | 0.014 |

**Table 6. Result of R2 and Q2.**

| Constructs | | R$^2$ | Q$^2$ |
|---|---|---|---|
| PER | 0.835 | 0.414 | |

calculated through bootstrapping [119]. The bootstrapping procedure was used to assess the significance and relevance of the structural model relationships, where empirical t-values were computed. If the t-value meets the threshold, the coefficient is considered statistically significant at a given significance level [119]. The results of the significance and relevance were showed in Table 5, while the R$^2$ and Q$^2$ values were displayed in Table 6.

In this study, five hypotheses were formulated to examine the relationships among the variables. As shown in Table 4, four relationships yielded t-values $\geq$ 2.33, indicating significance at the 0.01 level. Specifically, knowledge ($\beta$ = 0.188, p < 0.01), motives ($\beta$ = 0.173, p < 0.01), skills ($\beta$ = 0.526, p < 0.01), and traits ($\beta$ = 0.260, p < 0.01) positively affected job performance, explaining 83.5% of the variance in job performance. Therefore, H1, H2, H3, and H4 were supported. The R$^2$ value was 0.835, which exceeds the 0.26 benchmark, indicating a substantial model [131]. However, one hypothesis had a t-value below 2.33, and the Boot CI Bias Corrected interval included 0, suggesting no significant effect. Specifically, self-concept showed no significant relationship with job performance ($\beta$ = -0.100, t = 1.703, p>0.01), leading to the rejection of H5. Fig 2 displays the structural model result.

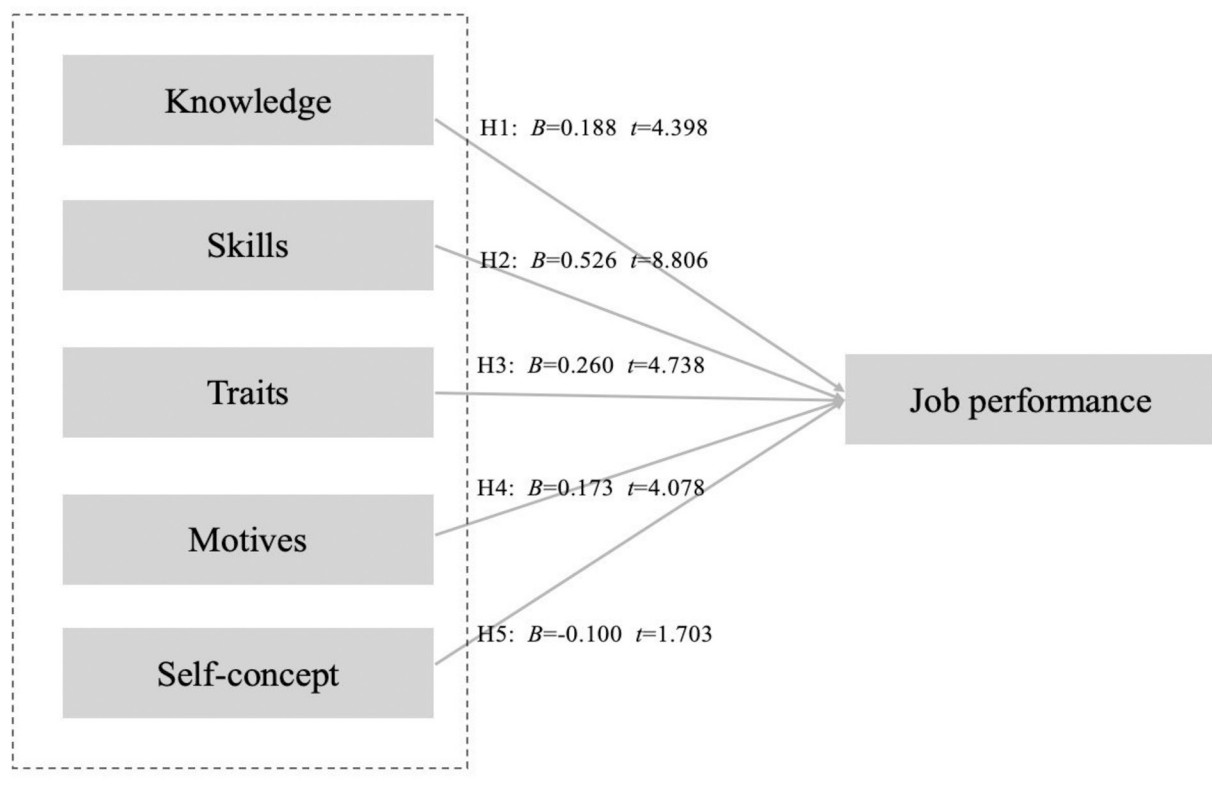

**Fig 2. Structural model result.**

## Discussion

The findings revealed that four work competencies—knowledge, skills, traits, and motives—were significantly related to job performance among university counsellors in Hunan province, while self-concept showed no significant relationship with job performance.

The current research demonstrated a significant relationship between knowledge and job performance (H1), consistent with studies by Rangchian et al. [72], Groza and Groza [73], Sujatha and Krishnaveni [74], and Liu and Guo [75]. According to Spencer's competency theory, knowledge can predict performance at work [34, 41, 104]. Martini et al. also concluded that improving work knowledge can significantly enhance the performance of the employee [132]. Professional knowledge is a crucial competency for counsellors, as it effectively predicts their individual performance [75, 133]. IThis is reflected in the study's demographic data, where 73.9% of counsellors held a Master's degree, and nearly 99% had a Bachelor's degree, indicating that most counsellors have extensive knowledge gained through education [134]. Additionally, university recruitment announcements often require at least a Bachelor's degree for counsellors [1], emphasizing the growing importance of knowledge in improving performance. These findings suggest that counsellors with higher levels of knowledge are more likely to achieve strong performance in their roles.

It was also hypothesized that skills would be significantly related to job performance among university counsellors. This study confirmed that skills had the strongest effect on job performance (H2), indicating that professional skills significantly enhance counsellors' performance. This finding aligns with previous research by Cho et al. [76], Moghimi et al. [79], and Jin and Song [135]. This finding is also in accord with the research of Negara [136], who stated that skill was highly correlated with individual performance, contributing 34.9 percent influence on performance. Skills can be cultivated through training, but mastering them often requires continuous practice and accumulation [136]. As Negara [136] emphasized, skills can be improved through practice and training, and this was also applied to counsellors as well. Thus, the finding implied that counsellors can improve their skills through practicing and training in future, which in turn can enhance individual performance.

As hypothesized, traits were significantly related to job performance among university counsellors. The current research confirmed this relationship (H3). The result showed that traits were positively related to job performance among university counsellors. This is in line with the previous studies by Oppong et al. [91] and Ghani et al. [92]. Zhang and Ziegler's longitudinal research highlighted that personality traits influence individual academic performance [137], while Tisu et al. found that personality traits can predict individual job performance [138]. This is also align with the research of Fichter et al. [139]. Specific traits such as enthusiasm, integrity, active listening, tolerance, affinity, personal charm, patience, care, love, and a sense of responsibility are essential for counsellors, as they help in managing complex student affairs and improving work effectiveness [47, 106, 140, 141]. The finding suggests that universities should focus on cultivating counsellors' traits to help them better adapt to the unique demands of their position and, ultimately, enhance their job performance.

Moreover, motives are often regarded as an important factor in improving job performance. This study confirmed that motives were significantly related to job performance (H4), supporting previous research by Sharma and Sharma [93], Zlate and Cucui [95] and Ren [96]. Gellatly et al. concluded that employees' motive was closely linked to their job performance [142]. Motivated employees had the incentive to perform and will always try to improve their performance [143]. Thus, the findings of this study are consistent with earlier research, showing that counsellors' motives positively influence job performance. Therefore, universities

should focus on addressing counsellors' needs, providing opportunities for promotion, and fostering a sense of belonging, which can effectively enhance overall performance.

It was initially assumed that once counsellors understood the social identities they needed to build in their roles—such as being a psychologist, educator, or life assistant—it would help them enhance their work efficiency. However, the results revealed that self-concept was not significantly related to job performance among university counsellors (H5). This unexpected finding contrasts with studies by Albert and Dahling [101], Khalaila [97], and Zanden et al. [103], all of whom emphasized that self-concept was significantly correlated with performance across various occupations. One possible reason was that the self-concept of the counsellor was diverse and vague. This also can be confirmed in the research of Wu, who stated that there existed a vague understanding of the role of counsellors in universities, and it was believed that all work involving students was the work of counsellors [144]. The diverse and complex nature of counsellors' roles may stem from misunderstandings or deviations in their self-concept. Another reason may be that part of the work of counselors may have high objectivity and standardization requirements. These jobs depend more on processes and systems than on counselors' self-concept, which may also lead to counselors' self-concept having no influence on job performance. Moreover, as counsellors accumulate professional skills and work experience, they may develop stable work methods and strategies. These methods may rely more on their professional knowledge and practical experience than on their self-concept. External factors, such as evolving student needs or changing policies, may also have a significant impact on counsellors' work. In such cases, job performance might depend more on how counsellors adapt to these external changes rather than their self-concept. Additionally, performance evaluations in universities tend to be based on objective metrics, meaning that counsellors' self-concept may not be a primary criterion in these assessments. Therefore, job performance may depend more on actual work outcomes than on counsellors' personal perceptions of their role.

In a word, this finding enriched the competency theory because this research verified whether the competency theory was still applicable to the position of university counsellor in China. The current research shows that some competency dimensions had unique relationships with job performance. Therefore, universities need to pay attention to improving counsellors' competence to improve their work performance and cultivate higher quality counsellors.

## Managerial implications

From a practical point of view, the results of the research can be used by the student affairs departments of universities that manage the construction of the counsellor team. The findings of this study showed that the work competencies components played an important role in enhancing counsellors' performance. Thus, in order to improve the performance of counsellors, more attention need to be paid to their competencies, such as knowledge, skills, traits and motives. It can be applied in the training, recruitment, career development and other links of the construction of the counsellor team.

Training is a vital tool for helping counsellors quickly adapt to their roles and address work deficiencies. Given the strong link between competencies and job performance, competency-based training is more focused and effective. Training programs should be designed with clear objectives aligned with key competencies such as knowledge, skills, traits, and motives. Appropriate targeted training programs can be formulated, and scientific and effective training methods can be selected to enhance training effects and promote counsellors' knowledge and skills. Specifically, the competency of counselors can be improved through different kinds of training, such as thematic training, practice and experience training, seminar training and

distance training, and vocational qualification and skill training. In view of the specific problems and challenges in counselors' work, organize special training, such as mental health education, career planning and employment guidance, etc., to enhance counselors' professional knowledge and skills in these fields. By analyzing real cases, role-playing and simulation training, counselors can experience coping strategies in different situations in simulated or real environments and improve their ability to deal with emergencies and solve practical problems. Seminar training can include group discussion and sharing, seminars and forums to promote exchanges and cooperation between counselors and broaden their horizons and ideas. Distance training can provide rich course resources by using the network platform, or realize collaboration and discussion among counselors across regions through remote collaboration tools such as video conferencing. Vocational qualification and skills certification training can improve professional knowledge and skills by holding professional qualification training such as psychological counselors. In addition, it can also organize corresponding skills training for specific skills needs of counselors, such as psychological counseling and employment guidance.

At the same time, the competencies of counsellors can also be used in the recruitment and selection of personnel because the results of this study showed that the competencies components of counsellors can benefit performance. Designing specific recruitment requirements around the competency content required by the position can make recruitment more clearly, which can truly select counsellors with core motivations, characteristics and high performance, and allocate appropriate personnel for the construction of the entire counsellor team. Counsellors' competencies include explicit features such as knowledge and skills, as well as implicit features such as traits and motives that can be considered in the selection criteria. In the job description, the competency requirements of counselors can be clearly listed, such as psychological counseling ability, and copy with emergencies. It can test candidates more comprehensively, making the selection of counselors more standard, rather than relying solely on the examiner's subjective impression and interview experience, which can also improve the overall quality of recruitment and finding the person fit for this job. In addition, evaluation methods such as scenario simulation and case analysis can also be introduced in the recruitment process to further test the practical operation ability of candidates.

Meanwhile, universities can establish corresponding performance appraisal indicators based on competency, which can better reflect the comprehensive performance of counsellors. The criteria should cover the core competencies of counselors, such as psychological counseling skills, student management skills, communication and coordination skills, crisis handling skills, etc., and clarify the specific performance and requirements. This performance appraisal system can allow counsellors with good job performance to be rewarded in time and improve the work enthusiasm of university counselors. For counsellors whose job performance is not ideal, it can help them improve their job performance through training or other means according to assessment standards and competency models, so as to meet the universities' expectations for counselors.

## Conclusion

This research investigated the relationship between work competencies and job performance among university counsellors. It was found that knowledge, skills, traits, and motives were significantly related to job performance except for self-concept. Therefore, university administrators must pay attention to improving counsellors' knowledge, skills, traits and motive rather than self-concept to improve job performance.

However, this study still has several limitations. The first limitation was this study relied on self-report measures to assess both work competencies and job performance. The use of self-report to measure both dependent and independent variables raises concern about the accuracy of causal conclusions due to various factors, such as systematic distortions in responses, common method variance, as well as the reliability and validity of the questionnaire scales' psychometric properties. Therefore, future research could use different methods such as interviews or objective data to collect multiple data. In addition, the cross-sectional nature of this study limits the ability to understand how the relationship between competencies and job performance might evolve over time. A longitudinal approach would allow researchers to track changes in these relationships and measure the long-term effects of interventions such as training on counsellors' competencies and performance. This would provide a clearer understanding of the dynamic nature of competencies and job performance. Moreover, the study focused on counsellors within the educational sector in Hunan province, which may limit the generalizability of the findings. This research showed that self-concept had no significant effect on job performance, while it may show a different result in other provinces. Therefore, future studies should explore the model in other fields, such as business, healthcare, or public service, as well as in different provinces, to verify whether the results hold across diverse settings. If future research is involved a comparative study, it may bring unexpected gains. Additionally, this research only examined the direct relationship between work competencies and job performance without considering potential mediating or moderating variables. Factors such as job satisfaction, professional identity, or work environment might play an important role in shaping how competencies impact job performance. Future research could explore these mediators and moderators to offer a more nuanced understanding of the relationships between these constructs.

## Supporting information

**S1 File.**
(XLSX)

## Author Contributions

**Conceptualization:** Jie Cao, Nur Naha Abu Mansor.

**Data curation:** Jie Cao, Jinhua Li.

**Investigation:** Jinhua Li.

**Supervision:** Nur Naha Abu Mansor.

**Writing – original draft:** Jie Cao.

**Writing – review & editing:** Nur Naha Abu Mansor, Jinhua Li.

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
