## [Decision Letter · Decision Letter 0]

1 Oct 2024

PONE-D-24-26338Impact of work competencies on job performance among university counsellorsPLOS ONE

Dear Dr. Li,

Thank you for submitting your manuscript to PLOS ONE. After careful consideration, we feel that it has merit but it needs some amendments. Therefore, we invite you to submit a revised version of the manuscript that addresses the points raised during the review process. Overall the study is well planned and appropriately excuted. 

We look forward to receiving your revised manuscript.

Kind regards,

Shazia Khalid, PhD

Academic Editor

PLOS ONE

Journal requirements: 1. When submitting your revision, we need you to address these additional requirements.Please ensure that your manuscript meets PLOS ONE's style requirements, including those for file naming. The PLOS ONE style templates can be found at https://journals.plos.org/plosone/s/file?id=wjVg/PLOSOne_formatting_sample_main_body.pdf and https://journals.plos.org/plosone/s/file?id=ba62/PLOSOne_formatting_sample_title_authors_affiliations.pdf 2. PLOS requires an ORCID iD for the corresponding author in Editorial Manager on papers submitted after December 6th, 2016. Please ensure that you have an ORCID iD and that it is validated in Editorial Manager. To do this, go to ‘Update my Information’ (in the upper left-hand corner of the main menu), and click on the Fetch/Validate link next to the ORCID field. This will take you to the ORCID site and allow you to create a new iD or authenticate a pre-existing iD in Editorial Manager. 3. Please ensure that you have specified a) Did participants provide their written or verbal informed consent to participate in this study?b) If consent was verbal, please explain i) why written consent was not obtained, ii) how you documented participant consent, and iii) whether the ethics committees/IRB approved this consent procedure." In consent please state in Ethics Method section and manuscript if it is written or verbal. If consent was verbal, please explain a) why written consent was not obtained, b) how you documented participant consent, and c) whether the ethics committees/IRB approved this consent procedure. 4. We note that you have indicated that there are restrictions to data sharing for this study. PLOS only allows data to be available upon request if there are legal or ethical restrictions on sharing data publicly. For more information on unacceptable data access restrictions, please see http://journals.plos.org/plosone/s/data-availability#loc-unacceptable-data-access-restrictions.  Before we proceed with your manuscript, please address the following prompts: a) If there are ethical or legal restrictions on sharing a de-identified data set, please explain them in detail (e.g., data contain potentially identifying or sensitive patient information, data are owned by a third-party organization, etc.) and who has imposed them (e.g., a Research Ethics Committee or Institutional Review Board, etc.). Please also provide contact information for a data access committee, ethics committee, or other institutional body to which data requests may be sent. b) If there are no restrictions, please upload the minimal anonymized data set necessary to replicate your study findings to a stable, public repository and provide us with the relevant URLs, DOIs, or accession numbers. For a list of recommended repositories, please seehttps://journals.plos.org/plosone/s/recommended-repositories. You also have the option of uploading the data as Supporting Information files, but we would recommend depositing data directly to a data repository if possible. We will update your Data Availability statement on your behalf to reflect the information you provide. 5. In the online submission form, you indicated that [Data will be made available on request.]. All PLOS journals now require all data underlying the findings described in their manuscript to be freely available to other researchers, either 1. In a public repository, 2. Within the manuscript itself, or 3. Uploaded as supplementary information.This policy applies to all data except where public deposition would breach compliance with the protocol approved by your research ethics board. If your data cannot be made publicly available for ethical or legal reasons (e.g., public availability would compromise patient privacy), please explain your reasons on resubmission and your exemption request will be escalated for approval.  6. Please include your full ethics statement in the ‘Methods’ section of your manuscript file. In your statement, please include the full name of the IRB or ethics committee who approved or waived your study, as well as whether or not you obtained informed written or verbal consent. If consent was waived for your study, please include this information in your statement as well. 

Reviewers' comments:

Reviewer's Responses to Questions

**Comments to the Author**

1. Is the manuscript technically sound, and do the data support the conclusions?

Reviewer #1: Yes

Reviewer #2: Yes

2. Has the statistical analysis been performed appropriately and rigorously? 

Reviewer #1: Yes

Reviewer #2: Yes

3. Have the authors made all data underlying the findings in their manuscript fully available?

Reviewer #1: Yes

Reviewer #2: No

4. Is the manuscript presented in an intelligible fashion and written in standard English?

Reviewer #1: Yes

Reviewer #2: No

5. Review Comments to the Author

Reviewer #1: The study makes a valuable contribution to the literature on work competencies and job performance, particularly in the context of university counsellors in China. The research design is sound and sample size and sampling technique was given and the findings are presented clearly and logically which is the good part. However, the paper could benefit more if the few recommendations would be added.

1. Add sampling characteristics also (you can add a table of demographic characteristics of sample, e.g. Age, gender, education, experience etc).

2. Clearly state the Spencer and Spencer description of the competency theory factors.

2. The need and process used for adaptation should be added before statistical models or analyses.

2. Add a more detailed exploration of the implications of its non-significant findings.

3. Add more critical discussion of its limitations.

Reviewer #2: The manuscript titled "Impact of Work Competencies on Job Performance among University Counsellors" clearly outlines the objective of the study, which is to investigate the relationship between various work competency factors (knowledge, skills, traits, motives, and self-concept) and job performance among university counsellors in China. The focus on this specific population, given the increasing emphasis on mental health support within educational settings.

However, the manuscript requires significant revision.

1. The document need editing corrections. There are many examples where vague and overly long sentences are used, leading to a lack of clarity.

2. The work competency factors are not operationally defined, which hinders the reader’s understanding of how these constructs are measured.

3. The tools used for data collection are not adequately discussed in the methodology section, that is critical to discuss.

4. The discussion seems to be focused on the conflicting roles of counsellors, which detracts from addressing the actual discussion according to the hypotheses.

5. Additionally, the suggestions for administrators and policymakers presented in the paper are overly broad. More specific recommendations would enhance the practical implications of the study.

6. Similarly, under the conclusion section, the suggestions are repeated. It would be more effective to suggest concrete and actionable recommendations.

6. PLOS authors have the option to publish the peer review history of their article (what does this mean?). If published, this will include your full peer review and any attached files.

Reviewer #1: No

Reviewer #2: No

---

## [Author Response · Author response to Decision Letter 0]

8 Nov 2024

Dear Editor and Reviewers,

Thank you for your valuable comments regarding our manuscript. We appreciate your insights, which have been instrumental in revising and enhancing our paper. We have carefully considered each point and made the necessary corrections. Below are our detailed responses to the reviewers’ comments:

Reviewer 1:

Reviewer point 1: Add sampling characteristics also (you can add a table of demographic characteristics of sample, e.g. Age, gender, education, experience etc).

Author response 1: The authors add the table 1 to show the demographic information of this research.

Reviewer point 2: Clearly state the Spencer and Spencer description of the competency theory factors.

Author response 2: The authors added more statements about the competency theory factors in theoretical foundation.

Reviewer point 3: The need and process used for adaptation should be added before statistical models or analyses. 

Author response 3: The author added the justification and process of choosing the statistical analysis method.

Reviewer point 4: Add a more detailed exploration of the implications of its non-significant findings.

Author response 4: The author added more explanation of the non-significant finding.

Reviewer point 5: Add more critical discussion of its limitations.

Author response 5: The author added more discussion of limitation part.

Reviewer 2:

Reviewer point 1: The document need editing corrections. There are many examples where vague and overly long sentences are used, leading to a lack of clarity.

Author response 1: The authors revised the full paper and finished the proofreading.

Reviewer point 2: The work competency factors are not operationally defined, which hinders the reader’s understanding of how these constructs are measured.

Author response 2: The definitions of work competency factors are added in the theoretical foundation section and in the literature review section.

Reviewer point 3: The tools used for data collection are not adequately discussed in the methodology section, that is critical to discuss.

Author response 3: The authors revised this part and added more statement about the

tools.

Reviewer point 4: The discussion seems to be focused on the conflicting roles of counsellors, which detracts from addressing the actual discussion according to the hypotheses.

Author response 4: To avoid the misunderstanding of this part, the authors revised the conflicting role and added more discussion about the result. 

Reviewer point 5: Additionally, the suggestions for administrators and policymakers presented in the paper are overly broad. More specific recommendations would enhance the practical implications of the study.

Author response 5: The authors added more specific recommendations in the managerial implication part.

Reviewer point 6: Similarly, under the conclusion section, the suggestions are repeated. It would be more effective to suggest concrete and actionable recommendations.

Author response 6: To avoid the repetition, the authors put the suggestion part inside the managerial implication part and give more specific recommendations.

---

## [Editor Report · Decision Letter 1]

27 Nov 2024

Impact of work competencies on job performance among university counsellors

PONE-D-24-26338R1

Dear Author,

We’re pleased to inform you that your manuscript has been judged scientifically suitable for publication and will be formally accepted for publication once it meets all outstanding technical requirements.

Kind regards,

Shazia Khalid, PhD

Academic Editor

PLOS ONE
---

## [Editor Report · Acceptance letter]

1 Dec 2024

PONE-D-24-26338R1 

PLOS ONE

Dear Dr. Li, 

I'm pleased to inform you that your manuscript has been deemed suitable for publication in PLOS ONE. Congratulations! Your manuscript is now being handed over to our production team.

Kind regards, 

on behalf of

Professor Shazia Khalid 

Academic Editor

PLOS ONE